# Enhancing Wire-EDM Performance with Zinc-Coated Brass Wire Electrode and Ultrasonic Vibration

**DOI:** 10.3390/mi14040862

**Published:** 2023-04-16

**Authors:** Apiwat Muttamara, Patittar Nakwong

**Affiliations:** 1Faculty of Engineering, Thammasat School of Engineering, Thammasat University, Khlong Luang 12120, Thailand; 2Faculty of Science and Technology, Department of Industrial Management Engineering, Phranakhon si Ayutthaya Rajabhat University, Ayutthaya 13000, Thailand

**Keywords:** wire-cut, ultrasonic vibration, zinc-coated brass, tungsten carbide

## Abstract

This study aimed to investigate the performance of zinc-coated brass wire in wire-cut electrical discharge machining (EDM) using an ultrasonic-assisted wire on tungsten carbide. The research focused on the effect of the wire electrode material on the material removal rate, surface roughness, and discharge waveform. Experimental results demonstrated that using ultrasonic vibration improved the material removal rate and reduced surface roughness compared to conventional wire-EDM. Cross-sectional SEM of the white layer and discharge waveform were investigated to explain the phenomena of ultrasonic vibration in the wire-cut EDM process.

## 1. Introduction

Wire Electrical Discharge Machining (WEDM) is a widely used process in the manufacturing industry for producing high-precision components in a variety of materials [1]. One of the challenging materials to machine with WEDM is superalloy due to its high hardness and strength. Sharma et al. investigated the effect of wire diameter on the surface integrity of Inconel 706, a material used in gas turbine applications, using WEDM. They also investigated the effect of wire material on productivity and surface integrity of WEDM-processed Inconel 706 for aircraft application [2,3]. One of the challenges for materials in WEDM is tungsten carbide. Many researchers have explored the use of wire-cut EDM to improve the cutting performance of tungsten [4,5,6,7,8,9,10]. Naveed et al. investigated the machining of curved profiles on tungsten carbide–cobalt composite using WEDM. They found that the machining parameters, such as wire tension and flushing pressure, significantly affect the machining accuracy and surface roughness of the machined surface [4]. Jangra et al. optimized the multi-machining characteristics in WEDM of WC-5.3% Co composite using an integrated approach of Taguchi, GRA, and entropy method [5,6]. They concluded that the proposed method can significantly improve the machining performance of WEDM. Muthuraman and Ramakrishnan optimized the WEDM parameters for WC-Co composites using the desirability approach [7]. They found that the optimal combination of machining parameters can significantly improve the MRR and surface finish of the machined surface. Masooth et al. investigated by modifying the Wire-cut electric discharge machining setup as turning process [8]. The results revealed that the pulse on time (T-on) and pulse off time (T-off) are more important than wire feed and wire tension. Shah et al. investigated that WEDM of tungsten carbide is influenced by various parameters such as pulse duration, voltage, and wire speed [9]. Parihar et al. investigated the effect of WEDM on the microstructure and mechanical properties of functionally graded cemented tungsten carbide. The study found that WEDM produced surface damage but the bulk FGCC microstructure is not affected by machining and the internal structure of the damaged layer had less WC grains as compared to top machined surface [10]. Many researchers compared the performance of wire-cut EDM with and without the zinc-coated brass wire electrode [11,12,13,14,15]. The results showed that using a zinc-coated brass wire electrode significantly improved the surface roughness and reduced the kerf width and geometrical accuracy compared to the conventional wire-cut EDM. This is because zinc-coated brass wire electrodes have been shown to provide better electrical conductivity and corrosion resistance compared to brass electrode materials. The coating is typically applied to a brass wire using heat treatment to ensure a tight attachment of the zinc layer to the wire surface. The zinc-coated brass wire has been found to produce smoother surfaces and reduce wire breakage during WEDM operations on tungsten [14]. Endo et al. investigated the instability and the machining efficiency is reduced as a result of poor flushing condition in a very small gap and observed the wire vibration applied to the workpiece, machining efficiency to improve lower discharge energy and feed rate such that EDM at a constant feed rate can further improve the machining efficiency [15]. Hirao et al. found that ultrasonic vibration-assisted EDM is expected to have a better effect on finish machining, especially when there is a small gap between the electrode and workpiece [16]. Similarly, Wansheng et al. and Gao et al. confirmed that vibration assistance in EDM improved both the material removal rate (MRR) and surface roughness [17,18]. Guo et al. [19] implemented ultrasonic vibration in wire-EDM and found a significant relationship between surface integrity and residual stresses. Their results indicated that surface roughness is relative to the discharge energy. Singh et al. developed ultrasonic vibration (UV) in EDM, which has the potential to enhance surface morphology with high accuracy and repeatability [20]. When vibration is applied to the micro-WEDM process, discharges are more effective with fewer short circuits [21]. Additionally, there exists an optimum relationship between vibration parameters, energy, and feed rate, such that EDM at a constant feed rate can further improve machining efficiency [22,23]. This study focused on the combination of ultrasonic vibration assistance and a zinc-coated brass wire electrode to evaluate their effects on the WEDM process for tungsten. The study will highlight the material removal rate, surface roughness, and discharge waveform during the process.

## 2. Materials and Methods

The wire-cut EDM machine used for the study is the Mitsubishi Wire-Cut System model FA-Advance series. Figure 1 shows photographs of the Wire-EDM machine. The schematic diagram of the ultrasonic setup for vibration in WEDM is shown in Figure 2, depicting two modes of vibration, namely vibrated wire and vibrated workpiece [18]. In this study, the vibrated wire configuration was chosen to investigate its efficacy for this purpose. The depiction of the ultrasonic arrangement for the wire and workpiece is presented in Figure 3. The vibration of the wire plays a crucial role in expelling and dislodging debris from the workpiece, further facilitated by the elevated pressure generated in the dielectric fluid. The amplitude of the wire increases with frequency, resulting in an increased number of node and antinode displacements. This higher frequency and amplitude of wire vibration also generates elevated pressure in the dielectric fluid, facilitating the expulsion of debris from the workpiece. The correlation between frequency, amplitude, and wire vibration is shown in Figure 4.

The experimental procedure began with the preparation of the workpiece and the setup of the WEDM machine with a zinc-coated brass wire and ultrasonic vibration. Table 1 shows element compositions of tungsten carbide. Figure 5 presents a schematic of zinc-coated brass wire and Table 2 shows CuZn37 alloy compositions for brass wire. Table 3 shows physical properties of tungsten carbide and Table 4 illustrates properties of zinc- coated brass wire. Parameters considered for the experiment included peak current, pulse duration, pulse off time, wire tension, and frequency of the ultrasonic vibration, with a selected frequency of 40 kHz [22]. The machine conditions are shown in Table 5.

## 3. Results and Discussion

### 3.1. Amplitude Measurment

To better comprehend the relationship between the amplitude of ultrasonic vibration and WEDM performance, it is essential to measure the amplitude during the machining process. Figure 6 presents the process of measuring the amplitude during the vibration-assisted WEDM process, where sensors were utilized to record the amplitude and frequency of both the tool terminals and the aluminum alloy transducer head. The vibration amplitude was measured at a frequency of 40 kHz, and the resulting displacement of the wire vibration is illustrated in Figure 7. The amplitude of the vibration obtained was found to range between 18 and 52 nm, as shown in the Figure 7.

### 3.2. Effect of Vibration Assistance on Materials Removal Rate and Surface Roughness

This comparison yields valuable insights into the performance differences between these two methods. When using the Wire-EDM machine, MRR is calculated by using the following equation:MRR = (Feed rate) × (wire diameter) × (work height)(1)
where ‘Feed rate’ in mm/min was measured during each trial, and wire diameter and work height in mm are constant. Surface Roughness number (Ra) was expressed in microns and the cut-off lengths were 2 mm. The number of samples used during the machining experiments varied depending on the specific conditions being tested. In general, experiments were carried out with at least three samples for each set of machining parameters to ensure the validity of the results. Figure 8 shows a comparison of material removal rate and surface roughness between USV-WEDM and conventional WEDM, yielding valuable insights into the performance differences between these two methods. The application of ultrasonic vibrations to the wire electrode via the transducer resulted in a significant increase in MRR, indicating the effectiveness of ultrasonic wire vibration mode. Furthermore, the use of ultrasonic vibrations also improved surface roughness. The benefits of ultrasonic-assisted vibration on materials removal rate in WEDM are due to several factors, including improved material removal rate by breaking down surface layers of the workpiece and making it easier for electrical discharge to remove the material. In addition, ultrasonic vibrations can increase the rate of thermal removal by promoting the breakdown of surface layers, which enhances material removal efficiency. Additionally, this phenomenon is also linked to the enhancement of surface roughness [17].

### 3.3. Scanning Electron Microscope (SEM) Analysis

The scanning electron microscope (SEM) image of the surface integrity of USV-WEDM is investigated to show differences compared to that of traditional WEDM. Figure 9 presents a SEM image comparing the surface integrity of USV-WEDM to that of traditional WEDM. The photograph shows a subtle variation in the impact of wire vibration on the wire surface, leading to a substantial enhancement in surface quality. The use of ultrasonic vibrations during the WEDM process can lead to the formation of finer spheroid features, as shown in the image. The analysis of the white layer formed during the WEDM process is crucial to understanding the surface finish quality. The white layer is a hardened and brittle layer that forms when the surface of the workpiece is heated to very high temperatures and melts before rapidly cooling down. Figure 10 illustrates the cross-sectional surfaces of the test specimen exhibiting the white layer of (a) vibration-assisted WEDM (USV-WEDM) and (b) traditional WEDM. The photograph clearly shows that the use of ultrasonic vibration during the WEDM process can lead to a thinner and more uniform surface with a reduced recast layer compared to traditional WEDM, resulting in improved surface finish. This is because the ultrasonic vibration helps to break down the surface layers and promote more thorough and efficient discharging, which results in less thermal damage to the workpiece and reduced recast layer formation [23,24,25].

Tungsten carbide is a highly sought-after material for its hardness and durability, but it is also known to be brittle, making it difficult to cut using traditional machining methods such as wire-cut electrical discharge machining (WEDM). The use of ultrasonic vibration in the WEDM process can cause the material to crack or break under the white layer due to several reasons. Firstly, high-frequency ultrasonic vibration generates surface stress on the tungsten carbide workpiece, which can lead to cracking or breakage. Secondly, the combination of ultrasonic vibration and electrical discharge in the WEDM process can generate high temperatures and rapid cooling cycles, leading to thermal shock on the surface of the tungsten carbide workpiece. Thirdly, tungsten carbide’s material properties, such as high hardness and low ductility, make it susceptible to breakage under high stress and high temperature conditions [26]. Finally, process parameters such as the vibration frequency, amplitude, and power, as well as the wire speed, discharge current, and pulse duration, can affect the material removal rate and the likelihood of breakage in the WEDM process. These factors must be considered when working with tungsten carbide to avoid unwanted cracks or breakages under the white layer.

The surface integrity of wire surfaces was examined using a Scanning Electron Microscope (SEM). Figure 11 illustrates the topography of the wire surface postprocess, with spherical particles representing melted material that solidified as globules and adhered to the wire surface.

The surface appears granular with uneven dispersion of spheroid features for normal WEDM; and short discharges are evident in area A, potentially due to insufficient flow of dielectric fluid causing inhomogeneous temperature distribution [17,18]. On the other hand, for the USV-WEDM, only a few fine spheroid features are observed, indicating a more thorough discharge process. Additionally, ultrasonic vibration may improve the stability of the zinc-coated brass wire, minimizing the risk of wire breakage and ensuring consistent cutting performance.

### 3.4. Waveform Analysis

To understand the benefits of USV-EDM, it is important to perform a waveform analysis of the process. This involves measuring and comparing the electrical waveforms generated by USV-EDM and traditional EDM wire cutting processes. By assessing the waveform characteristics, critical information can be obtained regarding the performance of the process and potential areas for improvement. Figure 12a depicts the voltage and current waveforms with ultrasonic vibration assistance, while Figure 12b shows the waveforms for conventional WEDM. Effective discharge represents the normal discharge generated at the set value. The number of effective discharges collected in 1 ms is illustrated in Figure 13, which demonstrates that ultrasonic-assisted WEDM generates more discharges than traditional WEDM. The effective discharge waveform in USV-EDM is more stable than in traditional EDM due to the removal of debris by ultrasonic vibrations. The ultrasonic vibration improves the stability of the discharge, reducing the risk of short circuits and resulting in more efficient material removal and better surface finish.

## 4. Conclusions

The application of ultrasonic vibrations to wire-cut electrical discharge machining (WEDM) has been demonstrated to have a significant impact on the material removal rate (MRR) and surface roughness (SR) of the workpiece. The use of ultrasonic vibration showed improved stability and increased MRR compared to a traditional WEDM. It was also noted that when the wire was vibrated, an effective discharge could be obtained more easily. The use of ultrasonic vibrations in WEDM has been shown to have a positive impact on machining performance, offering improved stability and better surface finish.

The limitation of our study is that only under specific experimental conditions was a limited range of ultrasonic vibration parameters considered. While the experiment settings led to improved machining performance and surface finish, there may be other combinations of parameters that could lead to even better results. Lastly, future studies could also focus on the development of advanced control systems for USV-EDM, which could improve the precision and repeatability of the machining process. This could involve the integration of advanced sensors and control algorithms to enable real-time monitoring and adjustment of the machining parameters.

## Figures and Tables

**Figure 1 micromachines-14-00862-f001:**
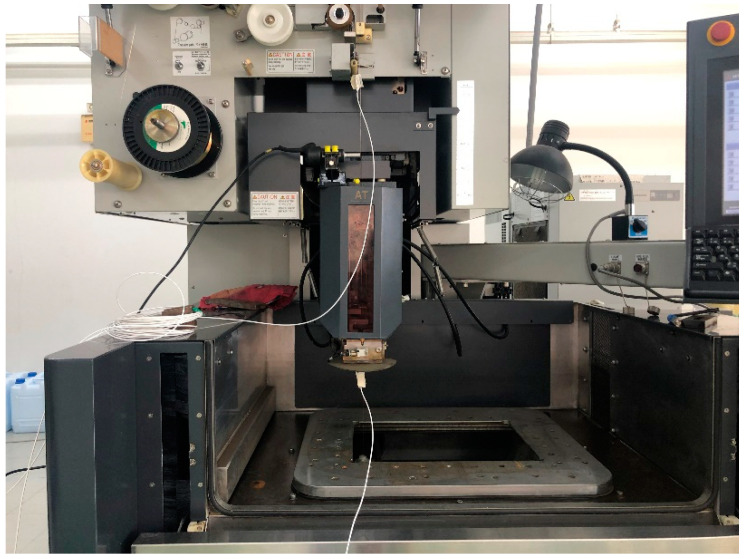
Wire-EDM machine.

**Figure 2 micromachines-14-00862-f002:**
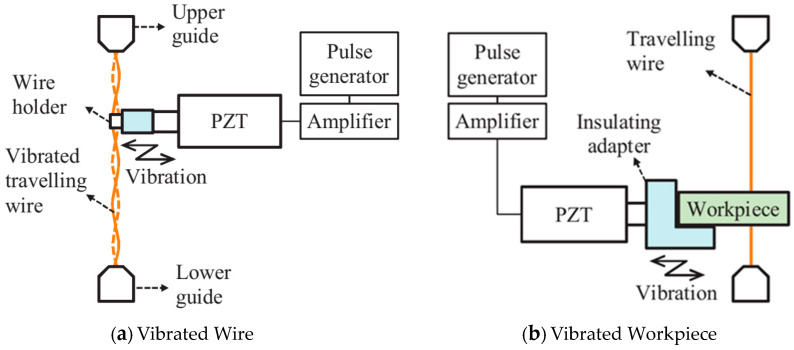
Schematic diagram of the ultrasonic setup for vibration in WEDM (**a**) wire and (**b**) workpiece [21]. Reproduced with permission from Hoang, K.T.; Yang, S.H., Mater. Process. Technol.; published by Elsevier, 2013.

**Figure 3 micromachines-14-00862-f003:**
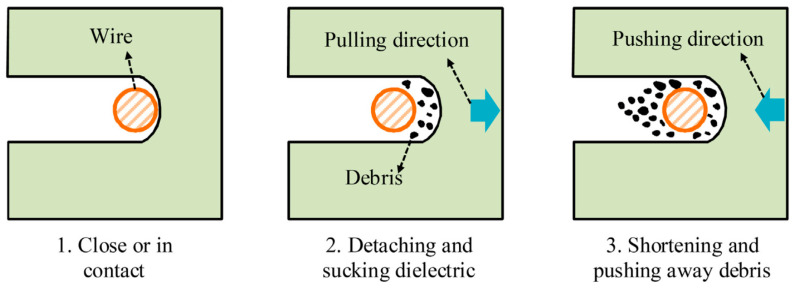
Principle of debris removal during ultrasonic vibration-assisted WEDM [21]. Reproduced with permission from Hoang, K.T.; Yang, S.H., Mater. Process. Technol.; published by Elsevier, 2013.

**Figure 4 micromachines-14-00862-f004:**
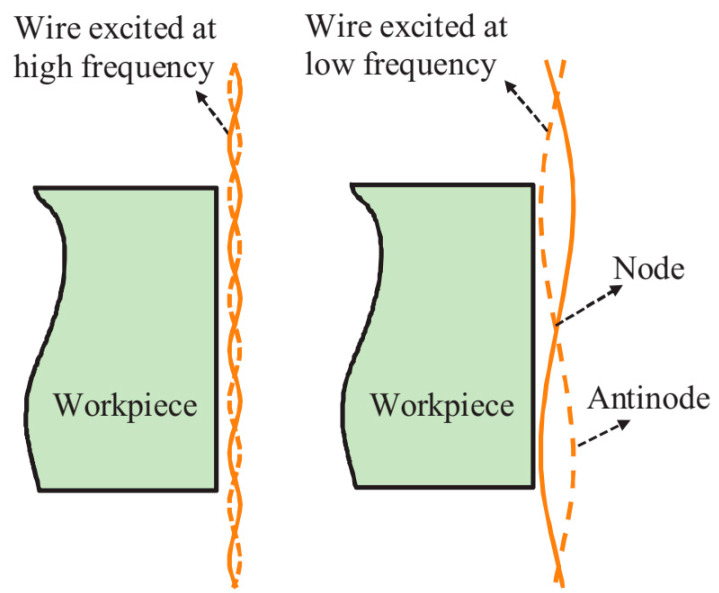
Relationship between wire vibration amplitude and frequency during ultrasonic vibration-assisted WEDM [21]. Reproduced with permission from Hoang, K.T.; Yang, S.H., Mater. Process. Technol.; published by Elsevier, 2013.

**Figure 5 micromachines-14-00862-f005:**
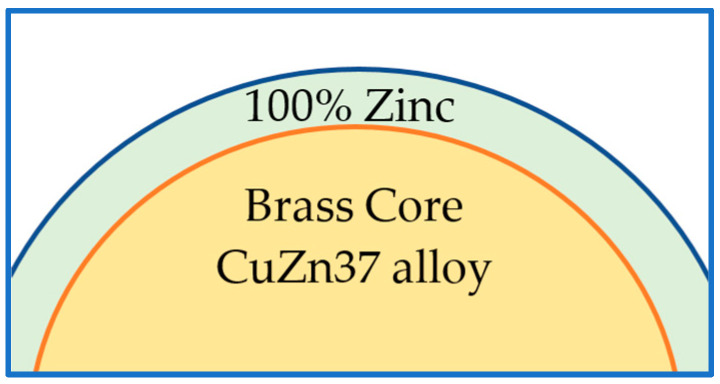
Schematic of zinc-coated brass wire.

**Figure 6 micromachines-14-00862-f006:**
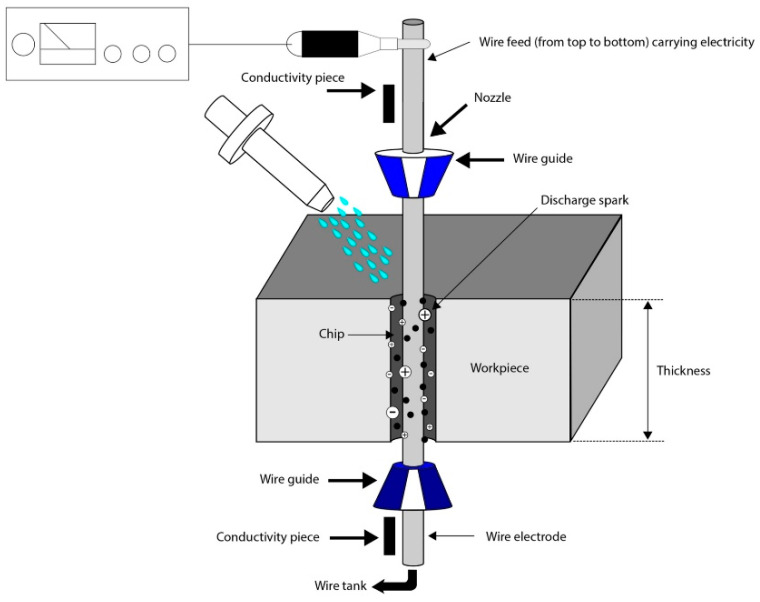
Measurement of Amplitude in Vibration-Assisted EDM Process.

**Figure 7 micromachines-14-00862-f007:**
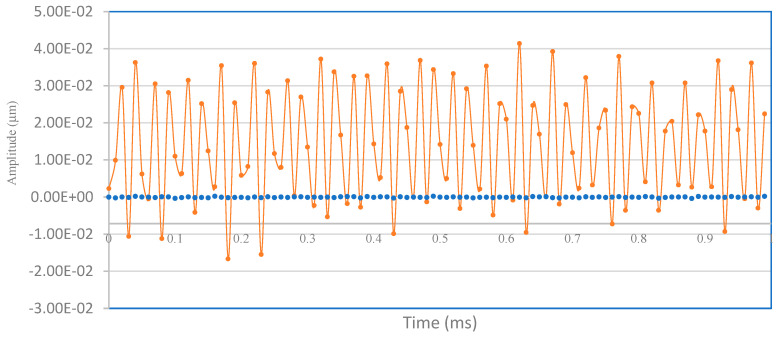
Wire vibration displacement measurements.

**Figure 8 micromachines-14-00862-f008:**
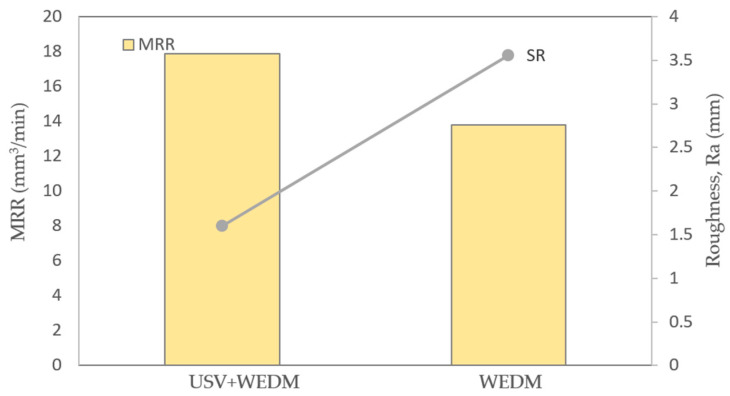
MRR and Surface Roughness for USV-WEDM and Normal WEDM.

**Figure 9 micromachines-14-00862-f009:**
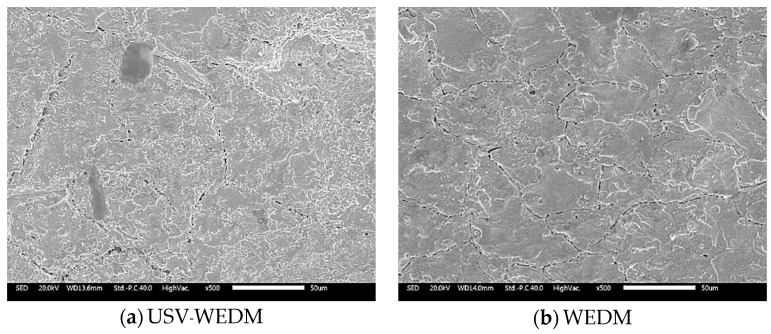
SEM images of surface of workpiece (**a**) with and (**b**) without Ultrasonic Vibration-Assisted WEDM.

**Figure 10 micromachines-14-00862-f010:**
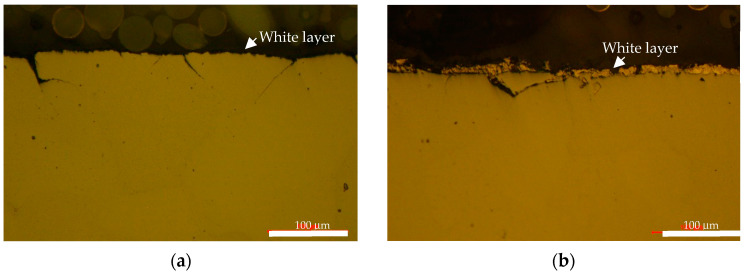
SEM photographs of white layer. (**a**) USV-WEDM; (**b**) WEDM.

**Figure 11 micromachines-14-00862-f011:**
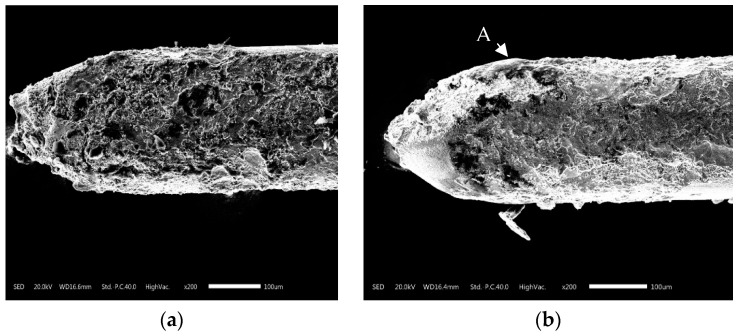
SEM photographs of wire surface. (**a**) Wire surface with USV-WEDM; (**b**) Wire surface with WEDM.

**Figure 12 micromachines-14-00862-f012:**
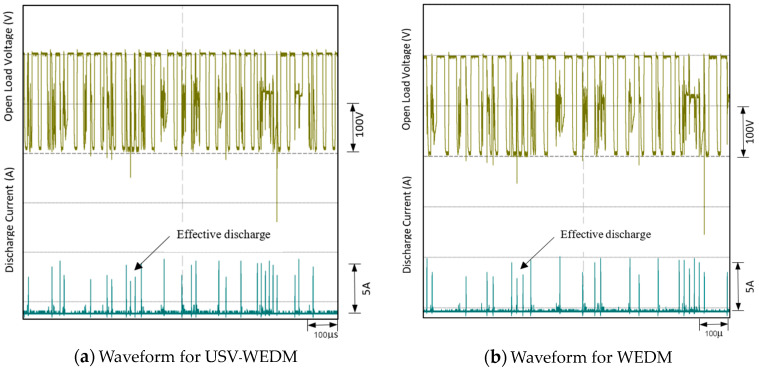
Discharge waveform for (**a**) USV-WEDM and (**b**) WEDM.

**Figure 13 micromachines-14-00862-f013:**
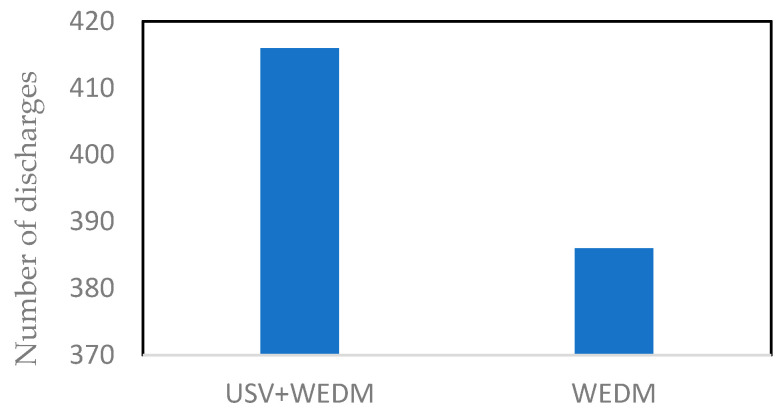
Number of effective discharges in 1 ms.

**Table 1 micromachines-14-00862-t001:** Element Compositions of Tungsten Carbide.

Element	W	Co	C	Ti
Weight %	73.98	9.18	11.41	5.43

**Table 2 micromachines-14-00862-t002:** Brass wire–CuZn37 alloy compositions.

Element	Cu	Al	Fe	Ni	Pb	Sn	Zn	Others Total
min	62.0	-	-	-	-	-	Rem.	-
max	64.0	0.05	0.1	0.3	0.1	0.1	-	0.1

**Table 3 micromachines-14-00862-t003:** Physical properties of tungsten carbide.

Melting Point (°C)	Density (g cm^−3^)	Thermal Expansion (°C)	Hardness (HRA)	Elastic Modulus (GPa)
2800	15.7	5 × 10^−6^	87.4	648

**Table 4 micromachines-14-00862-t004:** Properties of zinc-coated brass wire.

Standard Wire Diameter (mm)	Wire Diameter Tolerance (mm)	Tensile Strength (MPa)	Fracture Load (N)	Conductivity (%)
0.25	+/−0.001	883 (min)	43.3 (min)	20 (min)

**Table 5 micromachines-14-00862-t005:** Machining Conditions.

Voltage (V)	Current (A)	Wire Speed (m/min)	USM (kHz)
12	5	8	40

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
