# Peer review of "Enhancing Wire-EDM Performance with Zinc-Coated Brass Wire Electrode and Ultrasonic Vibration"

_micromachines, 2023, doi:10.3390/mi14040862_

Round 1
Reviewer 1 Report
Comments:
1. Manuscript have high scientific interest. Authors are advised to add few latest and related publications.
2. Once please go through the manuscript, in many places grammatical/ typo mistake are there.
3. Explore the introduction section by adding latest publications.
4. How many samples have been used during machining?
6. Add properties of tool and work material
7. Add the actual photographs of Wire EDM during machining.
8. Add surface roughness measurement conditions.
9. How MRR & SR are calculated, please describe in detail.
10. If possible add the actual photographs of work material where machining have been done.
Reviewer 2 Report
The author has well reported the enhanced Wire-EDM performance using a combination of zinc-coated brass electrodes and Ultrasonic vibration. However, the major corrections need to incorporate into the manuscript.
1. In Fig. 1, the author has demonstrated the schematic setup. The author is suggested to provide the actual image of the experimental setup to make it more appropriate for the reader.
2. Although the literature review is well elaborated, however, it can be enriched by considering a few more available articles in the relevant field.
https://doi.org/10.1016/j.jmapro.2016.09.001
https://doi.org/10.1016/j.precisioneng.2021.03.018
https://doi.org/10.1007/s11665-016-2216-z
3. In Table 1, the unit of voltage is wrong, the author is suggested to correct it.
4. In Figure 7, it would be better if SEM are compared at higher magnification (i.e., 1000X or 1500X 1) for better interpretation of WEDM and USV-WEDM.
5. In the conclusion section, the author mentioned that “The use of ultrasonic vibrations in WEDM has been shown to have a positive impact on machining performance, offering improved stability and better surface finish” but, the author has not produced any experimental data to claim better surface finish using ultrasonic vibration assisted WEDM. Justify?
6. The author is strongly advised to mention the future scope of the present study.
7. The author is strongly advised to mention the limitation of the current research work in the conclusion section.
Round 2
Reviewer 1 Report
Once please check gramaticle error throughout the manuscript.
Reviewer 2 Report
Since the author has successfully address all the comments raised by reviewer. Now, it can be consider for further processing.